

# Neighborhoods and adolescent polysubstance use in Jamaica

Amrita Gill[1], Erica Felker-Kantor[1], Colette Cunningham-Myrie[2], Lisa-Gaye Greene[3], Parris Lyew-Ayee[3], Uki Atkinson[4], Wendel Abel[2], Simon G. Anderson[5,6] and Katherine P. Theall[1,7]

[1] Department of Social, Behavioural and Population Sciences, Tulane University, New Orleans, LA, United States of America

[2] Department of Community Health and Psychiatry, University of the West Indies, Mona, Kingston, Jamaica

[3] Mona Geoinformatics Institute, University of the West Indies, Mona, Kingston, Jamaica

[4] Ministry of Health and Wellness, National Council on Drug Abuse, Kingston, Jamaica

[5] Faculty of Biology, Medicine and Health, University of Manchester, Manchester, United Kingdom

[6] The George Alleyne Chronic Disease Research Centre, Caribbean Institute of Health Research, University of the West Indies, Cave Hill, Barbados

[7] Department of Epidemiology, School of Public Health and Tropical Medicine, Tulane University, New Orleans, LA, United States of America

Corresponding author
Amrita Gill, agill6@tulane.edu

## ABSTRACT

**Background.** The purpose of this study was to identify latent classes of polysubstance use among adolescents in Jamaica and the role of neighborhood factors in the association with polysubstance use class membership.

**Methods.** This secondary analysis utilized a national cross-sectional household drug use survey conducted across 357 households in Jamaica (April 2016–July 2016) among a total of 4,625 individuals. A total of 750 adolescents (11–17 years) were included in this analysis. Latent class analysis (LCA) was conducted to identify polysubstance use patterns as well as latent neighborhood constructs. Neighborhood factors included social disorganization, concentrated disadvantage, community resources, community violence, and police station concentration. Multinomial regression analysis was implemented to evaluate the association between polysubstance use class membership and latent classes of neighborhood factors.

**Result.** The prevalence of lifetime polysubstance use was 27.56%. Four classes of polysubstance use were identified by comparing a series of five class models. The Bootstrap Likelihood Ratio Test (BLRT) indicated a good fit for the four-class model (< 0.001). The prevalence of alcohol latent classes was: (1) heavy alcohol users and experimental smokers (Class I) (15.20%), (2) most hazardous polysubstance users (Class II) (5.33%), (3) heavy smokers and moderate alcohol users (Class III) (7.07%), and (4) experimental alcohol users (Class IV) (72.44%). Three classes of neighborhoods were identified by comparing a series of four-class models. The prevalence of the neighborhood classes was: (1) low social disorganization and disadvantage (Class I) (58.93%), (2) high social disorganization and moderate disadvantage (Class II) (10.93%), and (3) high social disorganization related to perceived drug use and disadvantage (Class III) (30.13%). The BLRT indicated a good fit for the three-class model ($p = < 0.004$). Multinomial regression analysis indicated that adolescents living in neighborhoods with high disorder and moderate disadvantage (Class II) were 2.43 times (odds ratio (OR)) = 2.43, confidence interval (CI)) = 1.30–4.56) more
likely to be heavy alcohol users and experimental smokers (Class I) compared to experimental alcohol users, adjusting for sex, age, ethnicity, religion, and income. Class II of neighborhood classes presented with the highest levels of community violence (100%), perceived disorder crime (64.6%), police station concentration (6.7%), and community resources (low resources is 87.6%), while the concentrated disadvantage was moderate (14.8%).

**Conclusions**. Alcohol polysubstance use latent classes were identified among youth in this context. Neighborhoods with high disorder and moderate disadvantage (Class II) were associated with a higher likelihood of polysubstance use. The role of neighborhood conditions in shaping adolescent polysubstance use should be considered in policy, prevention, and treatment interventions.

## INTRODUCTION

Low- and middle-income countries (LMICs) rank 7th in disability-adjusted life years (DALYs) attributed to substance use and mental health among young adults (*Erskine et al., 2015*). Alcohol use is the largest cause of disease burden in young adults responsible for 10% of the global disease burden (*Degenhardt et al., 2016*). In the Caribbean, 8.4% of men and 3.2% of women between 15-19 years of age are involved in alcohol use; and 2.5% of men and 1.8% of women in the same age group are involved in illegal drug use (*Degenhardt et al., 2016*). While estimates vary by country and age, according to a school health survey in Jamaica, 64%, 27.5%, and 20.7% of students reported lifetime alcohol, tobacco, and marijuana use respectively (*Duncan, Duncan & Strycker, 2006*).

The effect of distal factors such as neighborhood conditions on adolescent substance use is well-studied in developed countries, while a few studies examine neighborhood factors in LMICs (*Byrnes et al., 2013*; *Razali & Kliewer, 2015*; *Skogen et al., 2019*). Some key neighborhood characteristics associated with substance use among youth in high-income countries include neighborhood disorder, social disadvantage, crime, abandoned buildings, and lower social cohesion (*Gentle-Genitty et al., 2017*; *Hadley-Ives et al., 2000*). Even among high income countries some studies show contrary results with no effect of neighborhood factors on adolescent substance use (*Jones & Adams, 2018*).

To our very best knowledge, we did not find any studies examining impact of neighborhoods on substance use in the Caribbeans. However, two studies identified the impact of neighborhoods on other outcomes such as violence (*Reyes et al., 2008*) and mental health (*Lowe et al., 2014*). For example, longitudinal study in the Caribbean context of San Juan, Puerto Rico identified social disorder as a predictor of violence in adolescents (*Reyes et al., 2008*). Varying profiles of neighborhood characteristics in four countries, the Bahamas, St. Vincent, St. Kitts, and Nevis, and Jamaica affected mental health outcomes differently with neighborhood factors more associated with depressive symptoms in Jamaican adolescents than in the other countries (*Lowe et al., 2014*).

Social disorganization theory is commonly used to describe the neighborhood-substance use relation (*Cleveland et al., 2010*; *Sampson, 2003*), which suggests that community organization is an important resource on which residents can draw to maintain supervision and control of youth. Disadvantaged communities suffer from a weak organizational base with a lower ability to engage in necessary informal social control that inhibits crime (*Haegerich et al., 2014*) and deviance (*Jiang & Dong, 2022*), including substance misuse. For example, economic hardship was associated with lower integration in neighborhoods and resulted in relational aggression among youth (*Jiang & Dong, 2022*). Similarly, neighborhood social processes such as neighborhood disorder, collective efficacy and neighborhood social capital were related to economic disadvantage and influenced youth violence (*Haegerich et al., 2014*). Lower levels of integration among communities has been related to collective efficacy and their ability to exercise social control against deviant activities such as violence and substance use (*Sampson, Morenoff & Earls, 1999*). Neighborhood social processes as predictors of polysubstance use latent classes in adolescents have rarely been studied.

Polysubstance use behavior is defined as the use of more than one substance either simultaneously or at a different time (*Connor et al., 2014*; *Davenport & Pardo, 2016*; *Duncan, Duncan & Strycker, 2006*). High prevalence of polysubstance use has been reported among vulnerable populations like transgender women and black cisgender sexual minority men (63.7%) and black sexual gender minorities (19.5%) (*Hotton et al., 2020*; *Moody et al., 2022*). Similarly, higher prevalence of substance use may be observed among vulnerable youth. Approximately 10.8% adolescents reported polysubstance use in three Caribbean countries of Jamaica, Trinidad Tobago, and Dominican Republic (*Peltzer & Pengpid, 2022*). While a higher prevalence of polysubstance use has been reported from Caribbean countries to our knowledge, no studies have examined latent classes of polysubstance use among adolescents in these contexts.

Latent class analysis (LCA) is a useful, person-centric approach that has been utilized to identify patterns of substance use and factors predicting them. LCA provides a deeper understanding of the risk profile of individuals by incorporating multiple dimensions of substance use like current use, past use, frequency of use, and combined use of substances (*Connell, Gilreath & Hansen, 2009*; *Gobel et al., 2016*). Various studies in the United States have examined polysubstance use among adolescents (*Min et al., 2022*; *Tomczyk, Isensee & Hanewinkel, 2016*) but only three studies to date, to our knowledge and at the time of this study, have reported neighborhood factors as predictors of latent classes of polysubstance use among adolescents.

These factors include community-mindedness (*Mitchell & Plunkett, 2000*), community availability of substances (*Connell, Gilreath & Hansen, 2009*), and community protection (*Cleveland et al., 2010*). Only one study among these was from a developing country (Argentina), the remaining two were conducted in the United States and Australia (*Tomczyk, Isensee & Hanewinkel, 2016*). This indicates a need to examine the role of neighborhood factors on patterns of polysubstance use behavior among adolescents in developing country contexts. We utilized LCA to not only classify substance use behavior among adolescents but also to classify neighborhoods based on objective and subjective

markers of social disorganization, structural determinants (disadvantage), and positive resources available within a community.

While fewer studies have examined the impact of latent classes of neighborhoods on child health outcomes (*Abner, 2014*; *McCoy et al., 2022*), none have examined the role of neighborhood latent constructs on substance use behaviors among adolescents. We included both subjective and objective markers of social disorganization to determine a latent neighborhood construct. Perceived disorder related to crimes, drugs, and alcohol were included as subjective markers of disorganization. Community violence determined by reports of crimes and the concentration of police stations in a neighborhood were included as objective markers of disorganization. While studies have implemented latent class analysis to classify neighborhoods according to levels of social disorganization (*Abner, 2014*; *McCoy et al., 2022*), to our knowledge, only one study included both the objective and subjective measures of crime and disorder (*McCoy et al., 2022*).

Crime and disorder have been seen to be higher in communities with greater concentrated disadvantage defined as a clustering of economic and social disadvantage within a community (*Jargowsky & Tursi, 2015*). For example, *Chamberlain & Hipp (2015)* found that neighborhoods with greater disadvantage are associated with greater crime rates. Concentrated disadvantage, in turn, may influence access to critical services such as good quality education and medical services within a community (*Jargowsky & Tursi, 2015*). We included objective measures of resources available in the community such as schools, churches, medical facilities, and other services such as business and personal services to determine positive resources.

Poor access to resources, concentrated disadvantage, and high crime and disorder may affect the abilities of communities to prevent harmful behaviors like polysubstance use among adolescents in Jamaica. Hence, the objectives of the present study add to the field by not only identifying latent classes of polysubstance use among adolescents in a developing country context but also identifying latent neighborhood constructs and examining their role in polysubstance use patterns among youth. Neighborhood level variables examined include markers of social disorganization, concentrated disadvantage, and community resources. We hypothesized that patterns of polysubstance use would be identified, and neighborhoods demonstrating high disorder, disadvantage, and low resources would be associated with a higher likelihood of polysubstance use.

## MATERIALS & METHODS

### Study setting and design

A national cross-sectional drug prevalence survey was conducted in Jamaica between April 2016–July 2016 (*Younger-Coleman et al., 2017*). The survey was aimed at determining the prevalence and pattern of drug use in the population 12-65 years of age. Multistage cluster sampling with enumeration districts (ED) as the primary sampling unit (PSU) was used. Parishes and area of residence were regarded as the first and second levels of stratification, respectively. Sixteen households were selected per ED. A random start was selected to identify the first dwelling to be selected. The final sample was 4,625 individuals, nested

within 313 EDs, considered a proxy for neighborhood and served as the definition of a neighborhood for the purpose of this study. EDs in the Jamaican national context have an average population of 482. The total sample of 4,625 was utilized for the construction of the neighborhood factors following which the data was subset to the adolescent population 12–19 years with a total of 750 adolescents included in the study.

Data were collected through in-person surveys. The survey was approved by the Ministry of National Security, Jamaica, and all subjects provided written informed consent. The current study, a secondary analysis, was approved by the University Hospital of the West Indies/University of the West Indies Ethics Committee.

## Measures

### Outcome- Substance use

The survey recorded the lifetime and current use of three substances tobacco, alcohol, and marijuana. We have included a total of seven binary measures of substance use behaviors:

*Tobacco use*  Two measures of ever and current tobacco use were constructed.

*Ever use of tobacco* was constructed from participant responses to the question: 'When was the first time that you smoked tobacco?' The response options were, 'Never', 'In the past 30 days', 'more than one month ago but less than one year ago'. Ever use was coded as 'No' for participant response of 'Never' and 'Yes' for the remaining responses.

*Current use of tobacco* was constructed from participant responses to the question: 'Do you currently smoke tobacco?' The response options were, 'Daily', 'Less than daily', 'Or not at all.' Current use was coded as 'Yes' for participant response, 'Smoked daily' and 'No' for the remaining responses.

*Alcohol use*  Three measures of ever, current use of alcohol, and binge drinking were constructed.

*Ever use of alcohol* was constructed from participant responses to questions: 'Have you ever drunk an alcoholic beverage.' Response options were, 'Yes' and 'No'.

*Current use of alcohol* was constructed from participant responses to the question: 'Have you drank alcoholic beverages in the past 30 days?' Response options were, 'Yes' and 'No.'

*Binge drinking* was constructed from the participant's response to the question: 'Over the past 2 weeks, how many times have you taken five drinks or more for males and four drinks or more for females on a single occasion/ outing.' Response options were, 'Not once', 'Just once', 'Two to three times, 'Between four and five times', 'More than five times.' These criteria for binge drinking have been determined by National Institute of Alcohol and Alcohol Abuse (*National Institutes of Health, 2004*). Binge drinking was coded as 'No' for participant response, 'Not once' and 'Yes' for the remaining responses.

*Marijuana use*  Two measures of ever and current use of marijuana were constructed.

*Ever use of marijuana* was constructed from participant responses to the question: 'When was the first time you smoked marijuana.' Response options were, 'Never', 'In the past 30 days', 'More than 1 month ago but less than 1 year ago', 'More than 1 year ago'. Ever use was coded as 'No' for participant response of 'Never' and 'Yes' for the remaining responses.

*Current use* was constructed from participant responses to the question: 'Have you smoked marijuana in the past 30 days. The response options were, 'Yes' and 'No'.

### Exposures-Neighborhood level variables

Neighborhood-level measures were constructed from both the household survey and objective neighborhood-level variables available from the Mona Geo-Informatics Institute (MGI) (https://main.monagis.com/). Two neighborhood-level measures were derived from the household survey. These include community alcohol use and community disorganization. The remaining were constructed from MGI.

Select neighborhood variables were also linked to participants' residential addresses and aggregated by ED, these included both survey responses aggregated to the ED level as well as secondary, objective data at the ED level. Objective data at the ED level for 2016 was obtained from MGI. All the neighborhood level variables were aggregated to ED level and z scores were created and standardized by dividing by population in each ED. The total sample of 4625 observations was utilized to aggregate the neighborhood variables as it is an appropriate level to estimate neighborhood-level perceptions and representations of the neighborhood structural elements examined in the study.

*Markers of social disorganization* Community disorganization was constructed from the participant's perception of select activities including drug dealing, breaking, and entering homes, scribbling graffiti on walls and damaging cables, taking drugs in public places, armed robberies or mugging, and young people loitering around street corners in their neighborhood, the methods are described in *Felker-Kantor et al. (2019)*. Specifically, the response to the questions: 'As far as you know, how much of the activities are in your neighborhood?' were included. Response options were on a four-point Likert scale ranging from 'One: a great deal' to 'Four: none' (*Felker-Kantor et al., 2019*). These were coded on a Likert scale of 'Zero-Three' as 'Zero' for responses 'Four: None' to 'Three' for 'One: a great deal.'

A total of seven variables of perception of select activities in respondents' neighborhoods were aggregated to calculate averages and z scores were created to standardize variables. Rates were created per thousand of the population per ED and factor analysis was implemented to reduce the data and create community disorganization scores. Two factors were identified with eigenvalues of 20.08 and 2.52, explaining 49% and 10% of the variability. Factor one loaded heavily on items related to drug dealing in the community, consuming drugs in the street, and young individuals loitering in the streets and was named community disorder drugs. Factor two loaded heavily on items related to perceived crime in the neighborhood including break-ins, armed robberies, and graffiti in the neighborhood and was named community disorder crime. The reliability of the Community Disorganization construct was also high at 0.90. Since the disorder variables demonstrated a highly skewed distribution a median split was implemented to dichotomize the variables as high disorder drug and crime, respectively.

*Community alcohol use* was constructed from participant responses to the question: 'Has a relative, friend or a doctor, or another health worker been concerned about your

drinking or suggested that you cut down?' The response options were on a 4-point Likert scale ranging from 'Zero: Never to 'Four: Daily or almost daily' Community alcohol disorder was coded as 'Zero' for participant response of 'Never' and 'One' for remaining responses 'One to Four'. Community alcohol disorder was aggregated to ED level and z scores were created and standardized by dividing by population in each ED. Since the community alcohol use demonstrated a highly skewed distribution a median split was utilized to dichotomize the variables as high community alcohol use.

*Community violence* was constructed from the number of crimes (murders, shootings, and robbery) per one thousand population in an ED. Since the community violence demonstrated a highly skewed distribution a median split was utilized to dichotomize the variables as high community violence.

*Police station concentration* was constructed from the number of police stations per one thousand population in an ED. Since the police presence demonstrated a highly skewed distribution a median split was utilized to dichotomize the variables as low police presence.

*Structural determinants  Concentrated disadvantage* was constructed from important demographic variables like population size, poverty, unemployment, distribution of males and females, and the age distribution in the population. Seven items were utilized to construct the concentrated disadvantage score. Five of these variables were constructed from MGI. These include percentage of population below poverty and counts of other demographic information: people with primary education, below 14 years and above 65 years of age, and unemployment. Two variables were included from the survey, the counts of female-headed households and unskilled workforce. These measures were aggregated to calculate averages and rates were created per thousand population of the ED. Z scores were created to standardize variables and factor analysis was implemented to reduce the data and create concentrated disadvantage scores. One factor with an eigenvalue of 6.02 was identified which explained 73.35% of the variability and moderate reliability at 0.78. Since the concentrated disadvantage score demonstrated a highly skewed distribution a median split was utilized to dichotomize the variables as high disadvantage.

*Positive resources  Community resources* were constructed from counts of seven types of resources available in the community. These include the number of schools, churches, gardens, businesses, grocery stores, general stores, and convenience stores available in the neighborhood. The seven measures were aggregated to calculate averages and rates were created per thousand of the population per ED. Z scores were created to standardize variables and factor analysis was implemented to reduce the data. One factor with an eigenvalue of 19.96 was identified for the Community resources which explained 55.8% of the variability and exhibited high reliability (Cronbach's alpha = 0.93). Since the community resources construct demonstrated a highly skewed distribution a median split was utilized to dichotomize the variables as low community resources.

## Data analysis

Data were reduced and managed in SAS version 9.4. Exploratory factor analysis was conducted to reduce the data and identify neighborhood factors. Internal consistency

reliability of the data-reduced index was also tested with Cronbach's alpha in SAS 9.4. Since no standard measures of neighborhood disorganization are available in the Jamaican context, three separate factor analysis were implemented to identify the neighborhood factors for concentrated disadvantage, community resources, and perceived community disorganization which have been detailed in the measures section.

After the neighborhood factors were identified the data were subset to the adolescent population aged 12-19 years. Latent class analysis was implemented to identify latent constructs of neighborhood factors and to identify patterns of polysubstance use in MPLUS software. Latent class analysis was chosen for its ability to identify unobserved classes across discrete or non-continuous variables. Criteria for a best-fit model included interpretability of the latent classes and fit statistics like Akaike Information Criteria (AIC) and Bayesian Information Criterion (BIC) (lower the score, the better the fit), as well as entropy value (closer to 1.0, the better the classification) (*Masyn, 2013*). Additionally, statistical model tests comparing k classes with k-1 classes including the Lo-Mendell-Rubin likelihood test (LMR) and Bootstrap Likelihood Ratio Test (BLRT) were implemented. Classes were added one at a time till the point no improvement in model fit was identified (*Masyn, 2013*). The LMR test utilizes derivatives from both k and k-1 class to estimate the two times the loglikelihood difference and model with best fit. The BLRT test not only estimates the k and k-1 class model to compute two times the loglikelihood difference but also repeats the step to several times to provide the true distribution of the difference (*Masyn, 2013*).

Class membership was predicted according to neighborhood-latent constructs identified and other individual-level factors through multinomial logistic regression in SAS 9.4 (SAS Institute, Cary, NC, USA). A two-step approach was followed. First, the crude estimates of the predictors on the alcohol latent classes were identified followed by adjusted estimates by including the significant predictors identified in the first step.

The data was described utilizing means and frequencies of socio-demographic variables, religious identity (others/Christian), ethnic identity (others/African origin/black), gender (men/women), age, income (low income/high/not known), substance use, and neighborhood measures included in the study.

## RESULTS

Study sample characteristics are presented in Table 1. The mean age of the study population was 16 years. Adolescent boys constituted 52.00% of the total study population, approximately 41.87% of the population belonged to urban areas and 20.10% had an annual income below USD 16.55. The religious and ethnic minority concentration in the population was low, at 17.73% and 4.67% respectively.

A high concentrated disadvantage was seen in 24.67% of the neighborhoods while low community resources were seen in 97.20% of the neighborhoods. High crime rates were reported in 27.07% of the neighborhoods while the perception of disorder crime and drugs were reported in 25.07% and 26.80% of the neighborhoods. A high concentration of police stations was reported in only 2.00% of the neighborhoods. High community alcohol disorder was reported in 28.67% of the neighborhoods.

**Table 1  Socio demographic and neighborhood characteristics of the adolescents in Jamaica (N = 750).**

|  | Percentages/Mean | Standard error | Minimum-Maximum |
|---|---|---|---|
| *Demographic characteristics* | | | |
| Percentage of non-Christian | 17.73 | – | – |
| Percentage of non-blacks | 4.67 | – | – |
| Percentage of males | 52.00 | – | – |
| Age | 15.89 | 2.23 | 12.00–19.00 |
| Income of households | | | |
| Percentage low-income | 20.13 | – | – |
| Percentage high-income | 59.33 | – | – |
| Percentage not known | 20.53 | – | – |
| Proportion Urban | 41.87 | – | – |
| *Neighborhood characteristics (%)* | | | |
| High concentrated disadvantage | 24.67 | – | – |
| Low community resources | 97.20 | – | – |
| High community disorder crime | 25.07 | – | – |
| High community disorder drug | 26.80 | – | – |
| High crime rate | 27.07 | – | – |
| High police rate | 2.00 | – | – |
| High community alcohol disorder | 28.67 | – | – |
| *Substance use (%)* tobacco ever use | 9.60 | – | – |
| Tobacco current use | 2.80 | – | – |
| Alcohol ever use | 53.47 | – | – |
| Alcohol Current Use | 22.93 | – | – |
| Alcohol binge drinking | 6.40 | – | – |
| Marijuana ever use | 12.40 | – | – |
| Marijuana current use | 6.53 | – | – |

A high prevalence of ever use (53.47%) and current use of alcohol (22.93%) was seen. Binge drinking was low at 6.40%. The ever use and current use of marijuana was 12.40% and 6.53% respectively. The ever use and current use of tobacco was 9.60% and 2.80% respectively.

## Latent classes of polysubstance use

A latent class analysis was performed, and parameters were estimated by utilizing the means of a likelihood ratio test (*Tekle, Gudicha & Vermunt, 2016*). As shown in Table 2 and Fig. 1, polysubstance use was seen in 27.56% of the adolescents. Approximately 15.20% of the adolescents were heavy alcohol users with high ever and current use of alcohol, some binge drinking, and experimental smokers with no current use of tobacco or marijuana but low ever use of tobacco and marijuana (Class I). Hazardous polysubstance use was reported by 5.33% of the adolescents (Class II). This group reported high current and ever use of tobacco, alcohol, and marijuana with heavy binge drinking. Heavy smoking and moderate alcohol use were seen in 7.07% of the adolescents (Class III). This group also reported

**Table 2  Prevalence of latent classes of adolescent polysubstance use behavior ($N = 750$).**

| | Polysubstance use latent class (Prevalence) | | | |
|---|---|---|---|---|
| *Adolescent substance use behaviour* | **Class I** (15.20%, $N = 114$) **Heavy alcohol users and experimental smokers** | **Class II** (5.33%, $N = 40$) **Most hazardous poly users** | **Class III** (7.07%, $N = 53$) **Heavy smokers and moderate alcohol users** | **Class IV** (72.44%, $N = 543$) **Experimental alcohol users** |
| *Ever use of tobacco* | 10.50 | 93.90 | 30.60 | 1.30 |
| *Current use of tobacco* | 0.00 | 44.70 | 4.30 | 0.00 |
| *Ever use of alcohol* | 100 | 100 | 87.60 | 36.80 |
| *Current use of alcohol* | 100 | 100 | 27.80 | 0.00 |
| *Binge drinking* | 15.30 | 74.60 | 0.00 | 0.00 |
| *Ever use of marijuana* | 3.70 | 100 | 100 | 0.00 |
| *Current use of marijuana* | 0.00 | 79.60 | 35.10 | 0.00 |

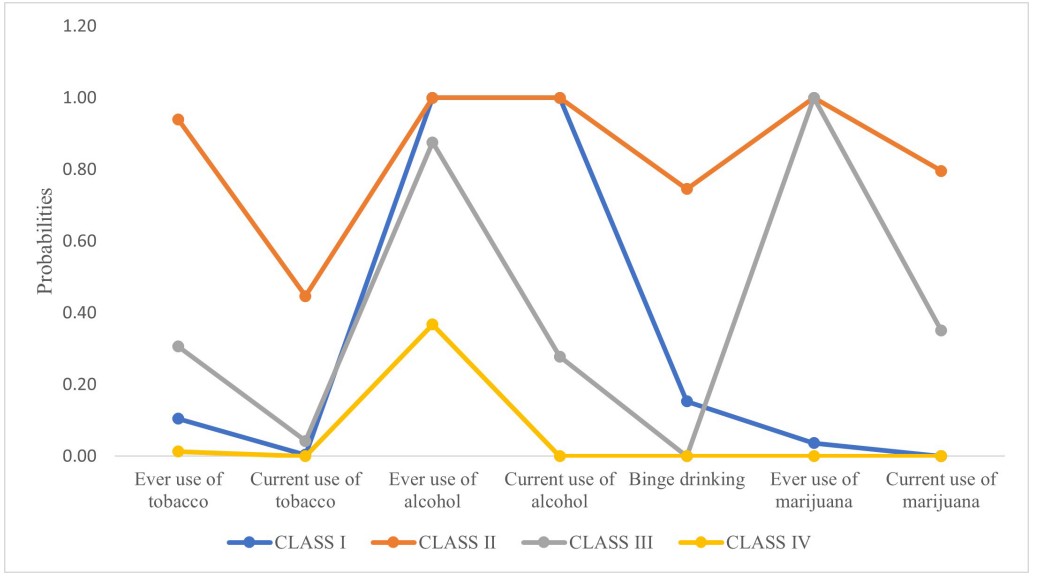

**Figure 1  Graphical representation of probabilities of latent classes of polysubstance use behavior.**

high ever use of alcohol, moderate ever use of tobacco and high current and ever use of marijuana.

The best fit model was the four-class solution, determined by comparing a series of five-class models. Indicators examined to demonstrate best fit includes degrees of freedom (96), AIC (2731.27), BIC (2876.12), entropy (0.99), LMR-LRT ($P < 0.001$), and BLRT ($P < 0.001$); coupled with model interpretability. The AIC, BIC, and SSBIC were the lowest for the four-class model indicating a good fit. While the LMR-LRT indicates that a five-class model was sufficient with a $p$-value of 0.05, the BLRT with a $p$-value of 0.07

**Table 3  Fit statistics for substance use latent classes.**

| Number of classes | Degrees of freedom | AIC | BIC | SSBIC | LMR-LRT | BLRT | Entropy |
|---|---|---|---|---|---|---|---|
| 1 | 118 | 4681.68 | 4714.02 | 4691.80 | – | – | – |
| 2 | 112 | 3026.53 | 3095.83 | 3048.20 | <0.001 | <0.001 | 0.91 |
| 3 | 104 | 2799.94 | 2906.20 | 2833.17 | <0.001 | <0.001 | 0.99 |
| *4* | *96* | *2731.27* | *2876.12* | *2874.50* | *<0.001* | *<0.001* | *0.99* |
| 5 | 88 | 2734.88 | 2915.06 | 2791.22 | 0.05 | 0.07 | 0.99 |

Notes.

AIC, Akaike Information Criteria; BIC, Bayesian Information Criteria; SSABIC, Sample Size Adjusted Bayesian Information Criteria; LMR-LRT, *p*-value for the Lo-Mendell-Rubin Likelihood Ratio Test; BLRT, *p*-value for Bootstrap Likelihood Ratio Test.

indicated a poor fit for the five-class model. Since the BLRT test is a better performing test (*Tekle, Gudicha & Vermunt, 2016*), the four-class solution was accepted (Table 3).

## Latent classes of neighborhood conditions

As shown in Table 4, Fig. 2, three latent classes of neighborhood variables were identified. All markers of social disorganization except community violence were low among Class I (58.93%). Community violence was moderately prevalent (20.00%). Concentrated disadvantage (4.70%) and community resources (low resources is 100%) were the lowest. Since most markers of social disorganization and concentrated disadvantage were low, Class I was named as low social disorganization and disadvantage and approximately 58.93% of the adolescents were included in this class. Class II (10.93%) presented with the highest levels of community violence (100%), perceived disorder crime (64.60%), and police station concentration (6.70%). The concentrated disadvantage was moderate (14.80%), and community resources were the highest (low resources is 87.60%). Class II was named as high social disorganization and moderate disadvantage and 10.93% of the adolescents were included in this class. Class III (30.13%) presented with a higher level of perceived disorder drugs (68.50%) than others and high concentrated disadvantage (64.40%) while the community resources were moderate (low resources is 95.20%). Community alcohol use was high with 42% prevalence in both Class II and Class III.

The best fit model was the three-class solution, determined by comparing a series of four-class models. Indicators examined to demonstrate best fit includes degrees of freedom (104), AIC (4448.32), BIC (4554.58), entropy (0.74), and LMR-LRT ($P = 0.003$) and BLRT ($P = 0.004$); coupled with model interpretability. The BIC was the lowest for the three-class model indicating a good fit. Additionally, the LMR-LRT and BLRT indicate that a three-class solution is adequate compared to a four-class solution (Table 5).

## Association between neighborhood conditions and polysubstance use

Multinomial logistic regression was performed to examine the bivariate association between neighborhood conditions, other socio-demographic factors, and polysubstance use (Table 6). Experimental alcohol users (72.44%) were used as a reference for polysubstance use behavior. According to maximum likelihood estimates, neighborhood class with high social disorganization and low concentrated disadvantage, ethnicity, religion, gender,

**Table 4 Prevalence of latent classes of neighborhood level variables (N = 750).**

| Neighbourhood level variables | Class I (58.93%, N = 442) Low social disorganization, disadvantage | Class II (10.93%, N = 82) High social disorganization, moderate disadvantage | Class III (30.13%, N = 226) High disorder drug, high disadvantage |
|---|---|---|---|
| Markers of social disorganization | | | |
| Community violence | 20.00 | 100 | 15.40 |
| Perceived disorder crime | 12.30 | 64.60 | 35.70 |
| Perceived disorder drug | 0.00 | 49.40 | 68.50 |
| High police station concentration | 0.70 | 6.70 | 2.80 |
| Community alcohol use | 19.1 | 42.00 | 41.90 |
| Structural determinants | | | |
| Concentrated disadvantage | 4.70 | 14.80 | 64.40 |
| Positive resources | | | |
| Low community resources | 100 | 87.60 | 95.20 |

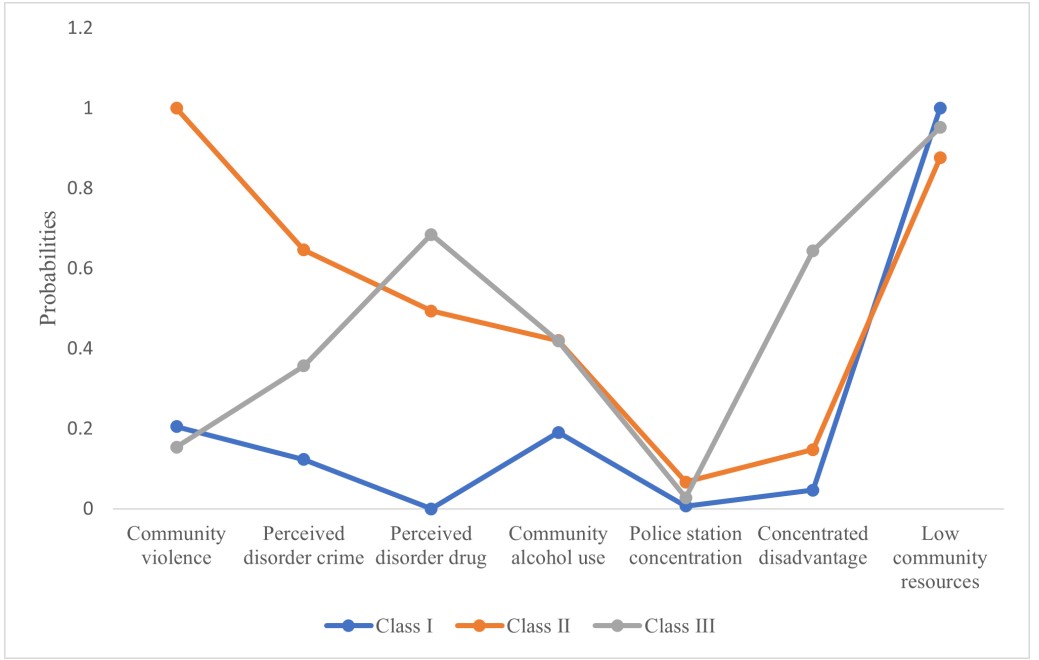

**Figure 2 Graphical representation of probabilities of Neighborhood latent classes.**

age, and income were associated with polysubstance use among adolescents in Jamaica. Adolescents living in neighborhoods with high social disorganization and moderate concentrated disadvantage were 2.34 times more likely to be heavy alcohol users and experimental smokers (odds ratio (OR)) = 2.34, confidence interval (CI)) =1.30−4.22) compared to experimental alcohol use. Males were 3.29 times more likely to be the most hazardous polysubstance users (OR = 3.29, CI =1.58−6.86) and 2.78 more likely to be

**Table 5  Fit statistics for neighborhood classes.**

| Number of classes | Degrees of freedom | AIC | BIC | SSBIC | LMR-LRT | BLRT | Entropy |
|---|---|---|---|---|---|---|---|
| 1 | 118 | 4681.68 | 4714.02 | 4691.80 | – | – | – |
| 2 | 112 | 4476.39 | 4545.69 | 4498.06 | <0.001 | <0.001 | 0.66 |
| *3* | *104* | *4448.32* | *4554.58* | *4481.55* | *0.003** | *0.004** | *0.74* |
| 4 | 96 | 4427.85 | 4571.07 | 4472.63 | 0.188 | *0.195* | 0.76 |

**Notes.**

AIC, Akaike Information Criteria; BIC, Bayesian Information Criteria; SSABIC, Sample Size Adjusted Bayesian Information Criteria; LMR-LRT, *p*-value for the Lo-Mendell-Rubin Likelihood Ratio Test; BLRT, *p*-value for Bootstrap Likelihood Ratio Test.

heavy smokers and moderate alcohol users (OR = 2.78, CI =1.49−5.17) compared to females. Christians were 1.88 times more likely to be heavy alcohol users and experimental smokers (OR =0.53, CI =0.33−0.87) and 2.28 times more likely to be heavy smokers and moderate alcohol users (OR =0.44, CI =0.23−0.83) compared to experimental alcohol use compared to non-Christians. One year increase in age was associated with a 1.38 times greater likelihood of heavy alcohol use with experimental smoking (OR =1.38, CI =1.24−1.54), 1.89 times greater likelihood of most hazardous polysubstance use (OR =1.89, CI =1.51−2.37), and 1.63 times greater likelihood of heavy smoking with moderate alcohol use (OR =1.63, CI =1.37−1.93).

Based on the results of the bivariate analysis, a multivariable multinomial logistic analysis was implemented including all variables that were significantly associated with polysubstance use (Table 7). While adjusting for sex, age, ethnicity, religion and income, adolescents living in neighborhoods with high social disorganization and moderate concentrated disadvantage were 2.43 times more likely to be heavy alcohol users and experimental smokers (OR = 2.43, CI =1.30−4.56) compared to experimental alcohol use.

## DISCUSSION

This study was among the first to examine the impact of neighborhood conditions on adolescent substance use in an LMIC, as well as to examine polysubstance use among youth in the Jamaican context. We observed a prevalence of polysubstance use based on alcohol, tobacco, and marijuana of 28%. While only a handful of national studies examine polysubstance use, this is comparable to other contexts, with the largest proportion of adolescents being non-users (*Tomczyk, Isensee & Hanewinkel, 2016*). In the U.S., reports of adolescent polysubstance use have been as high as 83% (*Cleveland et al., 2010*). We found a limited number of studies using latent class analysis for polysubstance use prevalence in a developing country context. Argentina and Iran, report the overall prevalence of polysubstance use up to 69% and 17.2% respectively (*Kabir et al., 2018*; *Pilatti et al., 2013*).

We also observed that neighborhood conditions may play a role in polysubstance use among adolescents in this sample. Neighborhoods with high disorder and moderate concentrated disadvantage were associated with polysubstance use with heavy alcohol use among adolescents. Most studies report higher substance use among adolescents exposed

**Table 6  Crude estimates of latent polysubstance use according to neighborhood and socio-demographic factors.**

| Adolescent substance use behavior | Class I (15.18%) Heavy alcohol users and experimental smokers OR (CI) | Class II (5.33%) Most hazardous poly users OR (CI) | Class III (7.06%) Heavy smokers and moderate alcohol users OR (CI) | Class IV (72.44%) Experimental alcohol users OR (CI) |
|---|---|---|---|---|
| **Neighbourhood classes** | | | | |
| Class 2 versus class 1 | 2.34** (1.30–4.22) | 1.69 (0.66–4.35) | 1.69 (0.70–4.09) | – |
| Class 3 versus class 1 | 1.27 (0.81–2.01) | 0.85 (0.40–1.83) | 1.33 (0.71–2.45) | – |
| **Ethnicity** | | | | |
| Non-blacks versus blacks | 2.98** (1.33–6.68) | 2.51 (0.70–8.95) | 3.22** (1.14–9.12) | – |
| **Religion** | | | | |
| Others versus Christian/Rastafarian | 0.53** (0.33–0.87) | 0.59 (0.24–1.10) | 0.44** (0.23–0.83) | – |
| **Gender** | | | | |
| Males versus females | 1.36 (0.90–2.03) | 3.29** (1.58–6.86) | 2.78** (1.49–5.20) | – |
| Age | 1.38** (1.24–1.54) | 1.89** (1.51–2.37) | 1.63** (1.37–1.93) | – |
| **Income** | | | | |
| Low versus high | 1.15 (0.69–1.93) | 1.13 (0.46–2.77) | 1.23 (0.61–2.49) | – |
| Unknown versus high | 1.31 (0.79–2.17) | 2.38** (1.15–4.90) | 1.18 (0.57–2.44) | – |
| Urban | 1.02 (0.68–1.54) | 1.46 (0.77–2.77) | 1.40 (0.80–2.47) | – |

**Notes.**

**Odds ratios that are significant.

⁻The referent category.

to social disorder. Studies in the U.S indicate a strong association between stress associated with community violence and substance use in adolescents (*Yule et al., 2000*; *Zinzow et al., 2009*). Among LMICs, a study in Puerto Rico observed an association between perception of violence/ social disorder and increased substance use in adolescents (*Reyes et al., 2008*). High alcohol use was reported among adolescents with low socio-economic status living in high disorder neighborhoods in Taiwan (*Chuang et al., 2007*). Similarly, in Malaysia, social disorganization has been associated with recreational drug use in adolescents (*Razali & Kliewer, 2015*). This study supports these findings.

The concentration of police stations was another marker of social disorganization in our study. The concentration of police stations was higher in neighborhoods with high social disorder and moderate disadvantage (Class II). Youth in these neighborhoods demonstrated heavy alcohol use and experimental smoking. While studies indicate a protective effect of police presence on substance use, we did not observe such an effect. For

**Table 7  Adjusted estimates of latent polysubstance use among adolescents according to neighbourhood latent construct ($N = 750$).**

| Adolescent substance use behavior | Class I (15.20%, $N = 114$) Heavy alcohol users and experimental smokers OR (CI) | Class II (5.33%, $N = 40$) Most hazardous poly users OR (CI) | Class III (7.07%, $N = 53$) Heavy smokers and moderate alcohol users OR (CI) | Class IV (72.44%, $N = 543$) Experimental alcohol users OR (CI) |
|---|---|---|---|---|
| Neighbourhood Class 2 versus class 1 | 2.43[**] (1.30–4.56) | 1.91 (0.69–5.33) | 1.79 (0.70–4.61) | – |
| Neighbourhood Class 3 versus class 1 | 1.43 (0.89–2.31) | 1.01 (0.45–2.29) | 1.68 (0.86–3.26) | – |
| Non-blacks versus blacks | 3.46[**] (1.54–8.24) | 3.08 (0.77–12.35) | 4.16[**] (1.35–12.35) | – |
| Others versus Christian/ Rastafarian | 0.61 (0.36–1.01) | 0.66 (0.29–1.48) | 0.53 (0.27–1.06) | – |
| Males versus females | 1.57[**] (1.02–2.4) | 4.24[**] (1.96–9.19) | 3.34[**] (1.84–6.81) | – |
| Age | 1.40[**] (1.26–1.56) | 1.98[**] (1.56–2.49) | 1.69[**] (1.41–2.01) | – |
| Income | | | | |
| Low versus high income | 1.09 (0.51–2.32) | 1.09 (0.51–2.32) | 1.26 (0.74–2.15) | – |
| Unknown versus high income | 1.26 (0.74–2.15) | 2.36[**] (1.08–5.16) | 1.12 (0.52–2.40) | – |

**Notes.**
[**]Odds ratios that are significant.
⁻The referent category.

example, the presence of police guards in schools has been negatively related to alcohol use and marijuana use (*Block, Swartz & Copenhaver, 2019*). Contrarily, within the LMIC context of Thailand, despite an increased police presence, drug use continued which challenged the role of police presence on drug use (*Werb et al., 2009*). This suggests that there may be other factors that affect the mechanism between police presence and drug use and the need to further explore this association in LMICs.

However, the prevalence of community disorder drugs was lower in the neighborhoods with a higher concentration of police stations. Another study reported the effectiveness of police in reducing the supply of drugs within communities. A higher police presence may crack down the drug markets and youth interaction with police may remove drug-related activities from public spaces to other markets (*Spooner, McPherson & Hall, 2004*). Hence the police presence may have shifted drug markets to other neighborhoods with lower police presence. We found a higher level of perceived community disorder related to drugs in neighborhoods with the lowest police presence and high disadvantage (Class III). However, we did not find an increase in substance use among youth in these neighborhoods, which requires further qualitative examination of the impact of police presence on substance use among communities in the Jamaican and LMIC context.

Contrary to our hypothesis, neighborhoods with higher community resources were associated with heavy alcohol use among youth. While we theorized community resources as an indicator of access to resources and improved social control or eyes on the street (*Aiyer et al., 2015*), such conditions may be an indicator of greater pedestrian traffic. According to previous research, mix-use neighborhoods with commercial businesses, convenience stores, bars, and even schools interspersed in residential areas may increase the number of active people on the streets increasing opportunities for crimes and other misdemeanors (*Browning & Jackson, 2013*). Taylor's territoriality model suggests that dense mix-use neighborhoods result in a greater inflow of outsiders in neighborhoods (*Taylor, 1988*). The presence of outsiders on the streets reduces territoriality or the tendency to maintain social order among residents (*Taylor, 1988*). Our risk classification of the neighborhoods for the community resources indicator is consistent with Taylor's territoriality theory.

Alternately, higher community resources may be a marker of urbanization. According to a multi-country analysis of alcohol and tobacco use, ever use of alcohol and tobacco and the age of onset of alcohol use were higher in urban youth compared to rural youth (*Mutumba & Schulenberg, 2019*). The area of residence was not significantly associated with polysubstance use membership in our study. Further analysis is required to assess the interaction effects between neighborhood classes and area of residence on polysubstance use.

While we had hypothesized that polysubstance use will be higher in neighborhoods with high concentrated disadvantage, our study indicates polysubstance use with heavy alcohol use and experimental tobacco use among adolescents in neighborhoods with moderate concentrated disadvantage. Similarly, other studies have shown lifetime rates of both alcohol and cigarette use higher in neighborhoods with greater social advantage (*Lyman & Luthar, 2014*; *Mutumba & Schulenberg, 2019*). While most studies from the U.S have shown concentrated disadvantage to affect adolescent delinquent behavior and substance use (*Kirk, 2010*) the contrary findings indicate further examination of inequalities within neighborhoods to examine an increase in substance use with greater social advantage. Even in our study, only neighborhoods with moderate concentrated disadvantage were associated with polysubstance use.

Strengths of this study are its novelty in terms of a polysubstance use latent class analysis in the Caribbean context and the inclusion of neighborhood latent constructs as predictors of classes of substance use. Also, the indices created to measure neighborhood factors were tested for reliability and validity and all demonstrated high validity and reliability in this context and should be utilized in future studies. Nonetheless, this study is not without its limitations, including its cross-sectional design and self-reported survey measures which may have introduced a degree of information bias. Furthermore, there are limitations to our definition of the neighborhood as an administrative boundary as well as with some of our neighborhood measures. No constructs of social capital were included, although research has also identified markers of social capital as a protective factor for adolescent risk behavior (*Shiell, Hawe & Kavanagh, 2018*; *Bjørnskov, 2006*). Also, a greater number of items were utilized for alcohol and tobacco use and fewer for marijuana use which might outweigh the importance of alcohol and tobacco use in the prediction of latent classes.

## CONCLUSIONS

Four latent classes of polysubstance use were identified among adolescents in Jamaica with 27.56% of youth reporting polysubstance use. As hypothesized neighborhood conditions predicted polysubstance use among adolescents in Jamaica. We hypothesized that neighborhoods with high social disorganization, concentrated disadvantage and low community resources would be associated with increased polysubstance use. While neighborhoods with high markers of social disorganization such as community violence, community disorder crime, high police presence predicted polysubstance use with heavy alcohol use and experimental tobacco use, the community disorder drugs, and concentrated disadvantage were low, but resources were high. Alternate theories might be operating in Jamaican neighborhoods to explain these findings. An explanation to lower community drug disorder despite a higher prevalence of other disorder types is the shifting of drug markets to other neighborhoods with lower police station concentration. An equity lens may help understand the lower concentrated disadvantage in highly disordered neighborhoods with heavy alcohol use and experimental smoking among youth. Similarly, higher resources among disordered neighborhoods may be explained by the models of territoriality where higher pedestrian traffic may diminish social control within neighborhoods.

Findings indicate the need to consider neighborhood conditions while designing interventions to prevent and treat polysubstance use. Context-specific studies to develop targeted interventions are required. Interventions should not only focus on more distal determinants to reduce inequalities, crime, and disorder in neighborhoods, but it might be effective to focus on a proximal determinant to improve territoriality within residents. Further, institutional mechanisms like policing may help reduce drug related disorder in communities.

## ACKNOWLEDGEMENTS

We acknowledge the Organization of American States, the Inter-American Drug Abuse Control Commission for support in revising the manuscript critically for intellectual content.

### Funding

The authors received no funding for this work, although Dr. Theall's Fulbright Scholar award aided in the development of neighborhood-level data. The funders had no role in study design, data collection and analysis, decision to publish, or preparation of the manuscript.

### Competing Interests

The authors declare there are no competing interests.

## Author Contributions

- Amrita Gill conceived and designed the experiments, performed the experiments, analyzed the data, prepared figures and/or tables, authored or reviewed drafts of the article, and approved the final draft.
- Erica Felker-Kantor conceived and designed the experiments, authored or reviewed drafts of the article, and approved the final draft.
- Colette Cunningham-Myrie conceived and designed the experiments, authored or reviewed drafts of the article, and approved the final draft.
- Lisa-Gaye Greene conceived and designed the experiments, authored or reviewed drafts of the article, and approved the final draft.
- Parris Lyew-Ayee conceived and designed the experiments, authored or reviewed drafts of the article, and approved the final draft.
- Uki Atkinson conceived and designed the experiments, authored or reviewed drafts of the article, and approved the final draft.
- Wendel Abel conceived and designed the experiments, authored or reviewed drafts of the article, and approved the final draft.
- Simon G. Anderson conceived and designed the experiments, authored or reviewed drafts of the article, and approved the final draft.
- Katherine P. Theall conceived and designed the experiments, authored or reviewed drafts of the article, and approved the final draft.

## Human Ethics

The following information was supplied relating to ethical approvals (i.e., approving body and any reference numbers):

This study was approved by The University of the West Indies, Mona, Jamaica.

## Data Availability

The data underlying this study contain potentially identifying sensitive participant information. Our study is a secondary analysis of the Jamaica National Drug Use Prevalence Survey 2016, the dataset for which, although de-identified by name and street addresses, utilizes geographical coordinates to link participants' residences to neighborhood-level measures on crime. We also collected other very sensitive information on substance use, both legal and illicit use. The dataset also includes several other direct and indirect identifiers that were necessary to conduct the analysis for the research.

Therefore, data can be made available only for use for academic purposes, and not for commercial purposes, on request.

Data requests can be made to the National Council on Drug Abuse, the Ministry of Health and Wellness Jamaica. Data requests received will be reviewed by the National Council on Drug Abuse on a case-by-case basis, considering the sensitivity of the data.

Data requests should include the name of the study, and be addressed to Mr. Michael Tucker, Executive Director, National Council on Drug Abuse, The Ministry of Health & Wellness Jamaica (mtucker@ncda.org.jm).

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
