# Peer review of "Neighborhoods and adolescent polysubstance use in Jamaica"

_PeerJ, doi:10.7717/peerj.14297_

## Round 0.1 · original submission · Major Revisions

Thank you for submitting the manuscript to PeerJ. It has been reviewed by experts in the field and we request that you make major revisions before it is processed further.

Please make the required changes, especially those of reviewer 2.

We look forward to hearing from you soon.

Best wishes,

Badicu Georgian, Ph.D

·

Basic reporting

We consider this research as original and appropriate with the aim and scope of the journal.
The article is written in clear English, respecting the technical and professional standard of redaction.
The references are relevant but relative actually for the topic of the article.

Experimental design

The research is focused on demonstrating the hypothesis and if the results and conclusion are relevant to this.
The study design and process of sampling and data collecting were clear and explicitly presented.

Validity of the findings

The results are well structured and conclusions are correlated with the hypothesis and highlight the novelty of this study, being a good start to extend the research about how the constructs of the neighborhood are related whit psychoactive substances use

Additional comments

R74-76, Maybe it will be useful to have more recent references about the topic.
R82-83 „Only three studies to date have reported neighborhood factors as predictors of latent 83 classes of polysubstance use among adolescents” – maybe is ok to mention all these studies.

It will be nice to find some more recent references related with article topic (eg:
Moody, R.L., Chen, YT., Schneider, J.A. et al. Polysubstance use in a community sample of Black cisgender sexual minority men and transgender women in Chicago during initial COVID-19 pandemic peak. Subst Abuse Treat Prev Policy 17, 4 (2022). https://doi.org/10.1186/s13011-022-00433-x
Vanderbruggen, N., Matthys, F., Van Laere, S., Zeeuws, D., Santermans, L., Van den Ameele, S., & Crunelle, C. L. (2020). Self-Reported Alcohol, Tobacco, and Cannabis Use during COVID-19 Lockdown Measures: Results from a Web-Based Survey. European addiction research, 26(6), 309–315.
Hotton, A. L., Balthazar, C., Jadwin-Cakmak, L., Gwiazdowski, B., Castillo, M., Harper, G. W., & Hosek, S. G. (2020). Socio-structural Factors Associated with Mental Health, Substance Use, and HIV Risk Among Black Sexual and Gender Minorities in the House and Ball Community. AIDS and Behavior, 1-8.)

Reviewer 2 ·

Basic reporting

The manuscript is well written and addressed a gap in the literature and addresses potential factors that directly impact adolescent polysubstance use. The manuscript makes a nice contribution to the understanding of the effect of neighborhoods on adolescent polysubstance use in Jamaica.

Please see several suggestions mentioned for each section below.

Experimental design

1) The authors reported that data were collected through interviews. However, it is confusing that authors used "the survey" after "face-to-face interview".
2) Lines 129-133: Binge drinking was measured as a response to the number of times five drinks or more for males and four drinks or more for females were consumed on a single occasion in the last two weeks
(not once, just once, 2 to 3 times, between 4 and 5 times, more than 5 times) and recoded to
create a variable of frequency of binge drinking (not once, just once, two-three times, four or
more times). Why is the classification of binge drinking is differ by gender? Do you have a reference or rationale for it?
3) The authors may add a sentence describing how many items were used to measure each construct at the beginning of each measure. That would be very helpful for readers to read each measure efficiently.

Validity of the findings

1) Table 1 shows the results of factor analysis but in the text, the authors described sample characteristics. It is better to change the order of Table 1 and Table 2.
2) The authors need to add a general description of demographic characteristics of the latent classes (e.g., gender, age, first-generation to go to college, income, disadvantaged background and etc.). The authors included the important information in Table 2. Why don't the authors look at the differences in this information by latent classes? Are there any demographic differences/geographical differences among latent classes? Table 3 should have that information too.
3) The authors need to include the number of each class next to % in Table 4 and compare the number of classes using LRT and BLRT (see https://doi.org/10.1016/j.ypmed.2019.01.012).
4) The authors may add a Figure that depicts the latent class profile (see https://doi.org/10.1016/j.jsat.2015.01.012).
5) The authors did not explain why the sample sizes used in the study differed between Table 1 and the other Tables.
6) Have the authors thought about making a latent factor (construct) of "neighborhood" to predict latent classes of adolescent polysubstance use? All neighborhood factors may be highly related. If does, the authors may include correlations among latent factors.
7) Table 1 is confusing. EFA was conducted for each construct. However, Table 1 looks like EFA was conducted including all neighborhood variables. The authors need to clarify the description of variables in Data Analysis too.
8) The authors need to explain why crude estimates of the odds ratio were more appropriate than estimates of the adjusted odds ratio. If neighborhood variables are related to each other, interpretation of the odds ratio for each covariate variable by holding all other variables within a model could be better.

Additional comments

1) Lines 82-86: The authors mentioned "Only three studies to date have reported neighborhood factors as predictors of latent classes of polysubstance use among adolescents. These factors include community-mindedness, community availability of substances, and community protection. Only one study among these was from a developing country (Argentina), the rest having been conducted in the United States
(18) and Australia (2) (Tomczyk et al., 2016)." : What do the numbers, (18) and (2), represent? Do these numbers represent the number of studies conducted in each country? The authors mentioned that only three studies have reported neighborhood factors right before this sentence.
2) Did the authors use social disorganization theory for the current study? A detailed explanation of this theory is needed in the introduction.
3) Add references for AIC and BIC. Before saying "AIC" and "BIC", the authors need to explain what it is.
4) The references need to be revised based on APA 7 (e.g., need doi for each reference).
5) "chi-square" should be replaced by χ2.

---

## Round 0.2 · Minor Revisions

ABSTRACT: There is a lack of numerical information, with statistical representation, to understand the findings of the study. In this way, presenting the difference values with statistical significance is important for a better understanding of the reader.

---

## Round 0.3 · accepted · Accept

Accepted for publication. Congratulations!